# Autobidding with Interdependent Values

## Abstract

In this paper, we initiate the study of autobidding where the signals for each bidder can be noisy and correlated. Our first set of results showcases the failure of traditional auctions such as the second-price auction (SPA) and the first-price auction (FPA). In particular, uniform bidding is not an optimal bidding strategy for SPA and both SPA and FPA can have arbitrarily poor efficiency. To circumvent this, we propose the Contextual Second Price Auction (CSPA), a novel mechanism which mitigates the aforementioned adverse effects by leveraging multiple signals to adjust the allocation of SPA. We show that uniform bidding is an optimal bidding strategy in CSPA and we prove a tight bound on the price for anarchy for CSPA of 2, thus recovering the well-established results in the independent setting. Finally, we show that CSPA always achieves at least half the welfare of SPA; moreover this is also tight.

## CCS Concepts

• **Applied computing** → **Electronic commerce**; **Online auctions**; • **Theory of computation** → *Theory and algorithms for application domains.*

## Keywords

Autobidding, Uncertainty, Winner's Curse, Interdependent Values

**ACM Reference Format:**
Anonymous Author(s). 2025. Autobidding with Interdependent Values. In *Proceedings of ACM Web Conference 2025 (WWW '25)*. ACM, New York, NY, USA, 9 pages. https://doi.org/XXXXXXX.XXXXXXX

## 1 Introduction

Recent advancements in machine learning have improved online advertising auctions, empowering autobidding systems to leverage granular signals (e.g., clickthrough rates, conversion rates) for each impression to set bids on behalf of the advertiser. However, such granularity can potentially introduce two critical challenges that further complicate the bidding problem:

(1) **Inherent Prediction Errors:** Noise in the predictions, coupled with the winner selection mechanism, increases the likelihood that winning bids are driven by overestimation.

(2) **Signal Correlation:** Shared user characteristics across advertisers lead to a selection problem, often called the "winner's curse": the winning bidder is likely to have a larger

signal compared to other bidders, and hence, likely to overestimate the value of the query.[1]

To mitigate this problem, traditional auction theory literature has shown that open auctions, like an ascending auction, have good properties to alleviate the winner's curse problem when bidders have profit-maximizing goals [24]. Modern ad auction bidders however adopt automated bidders, or autobidders, to submit their preferences in online auctions [3]. Unlike profit-maximizing bidders, the objective of these autobidders is to optimize for a campaign's total value subject to high-level constraints such as a return-on-spend (ROS) constraint. How to design an auction that is robust to the winner's curse problem when the bidder's goal is to maximize value, instead of profit, remains an open question.

In this paper, we provide a first step towards understanding the problem of the winner's curse in the autobidding world. We base our study in a theoretical model where autobidders' valuations are private but correlated across the advertisers. There are $n$ bidders interested in buying queries owned by a platform (the auctioneer). We consider a Bayesian query-model so that the measure (or, probability) of queries where bidders' valuations are $\mathbf{v} = (v_1, \ldots, v_n)$ follows a distribution $F$.[2] Bidders have an autobidding objective and their goal is to maximize the expected value they obtain across impressions subject to a return on spend constraint. We depart from the canonical autobidding models and assume that the value per query $v_i$ is unknown to everyone and each bidder, using a machine learning model, predicts an unbiased signal $\tilde{v}_i$ of $v_i$. Importantly, we assume that a bidder $j$'s signal $\tilde{v}_j$ may be correlated with bidder $i$'s unknown value $v_i$. For example, the correlation may be due to features that are common to all autobidders, such as user features. In particular, this means that the inference made from bidder $j$'s prediction model may be valuable information to bidder $i$. We will refer to this as the interdependent value setting.

Our first set of result explores how classic auction results in the autobidding literature change in the interdependent value setting. We show that the second-price auction (SPA) does not retain the appealing property that the optimal bidding strategy is a simple multiplier on top of the prediction $\tilde{v}_i$, known as a uniform bidding property [2]. The reason is that, in an environment with interdependent values, the runner-up's bid now has two effects: in addition to the *pricing* effect, which is due to the second-price nature of the auction, it has an *information* effect since it affects the winning bidder's assessment of the value of the query. A bidding policy that accounts for the distance between the runner-up bid and the winning bid can perform better than uniform bidding (Theorem 5.1).

We then study the welfare implications of running SPA when the values may be interdependent. We focus on a worst-case study using the standard price of anarchy metric (PoA), and show that there are some instances where the welfare of SPA can be arbitrarily

---

[1]Mehta and Lee [23] shows empirical evidence of the winner's curse effect on electronic auctions.

[2]Even though our baseline model is set as Bayesian model, this can be equivalently represented as the continuous query model studied in Alimohammadi et al. [4], Perlroth and Mehta [25].

worse than the optimal allocation. In other words, for SPA we have PoA = ∞ (Theorem 5.3). This is in stark contrast to the previous result that PoA = 2 for the case with independent valuations [2].

Next, we turn our attention to the case where the auction is a first-price auction (FPA). While it is known that with independent valuations uniform bidding is suboptimal, research has shown strong efficiency properties when autobidders are constrained to a uniform bidding strategy on FPA. In particular, it is an equilibrium of the constrained game for bidders to bid their value. The allocation is always efficient (PoA = 1) [11]. We, however, show that neither property holds when valuations are interdependent: bidders (sometimes) shade their value (Theorem 5.2) and we show that, in the worst case, we can have PoA = ∞ (Theorem 5.3).[3]

Given the negative results of FPA and SPA, in Section 6 we introduce a new mechanism called Contextual SPA (CSPA). In CSPA, the platform weighs each bidder's bid according to a contextual, collective estimate of the common user type. The platform effectively aids bidders' posterior predictions by constructing an intermediate, more accurate signal that captures the information contained in each impression separately.[4] The winner and its price is computed following a SPA, but the bidders' bids now receive contextual information, and is thus a more precise estimate of the value of that impression.

Our mechanism is inspired by the concept of open auctions. Examples of this include (i) the English auction where an auctioneer announces a sequence of increasing bids and a bidder is able to observe if there exists a bidder that is willing to accept the bid and (ii) the Japanese auction where a public price is announced and the bidders are able to observe the set of bidders that remain in the auction. Both of these auctions end when there is only a single bidder remaining. The important property of open auctions is that, as the auction progresses, information about the other bidders' is implicitly revealed. This allows bidders to update their priors at each round and, under some natural assumption on the bidder's priors, such auctions are effective at mitigating the winner's curse problem when bidders have profit-maximizing objectives, as shown in Milgrom and Weber [24]. Along these lines, CSPA can be viewed as an extreme form of information revelation where all information is revealed prior to the auction instead of incrementally.

We then show that the CSPA restores the optimality of uniform bidding (Theorem 6.1). The reason is that by adding the contextual information at auction time, CSPA is able to remove the information effect problem on the bidder's decision and, thus, making the bidding problem look like a SPA problem with independent values. The next result measures the impact on welfare of CSPA. Compared to the optimal allocation, we show that PoA = 2 (Theorem 6.2), thus again recovering this positive property of SPA under independent values. We also compare the welfare outcome relative to SPA. We show that CSPA also achieves at least half of the welfare that is achievable by SPA but the welfare achievable by CSPA can be arbitrarily more than the welfare of SPA (Theorem 6.3). In fact, these results are tight.

## 1.1 Contributions

To the best of our knowledge this is the first paper studying the problem of interdependent values in auctions in the context of autobidding. As we show in the paper, having interdependency between the bidders' valuations can dramatically change well-established results in auction design with autobidders. Our results demonstrate that it can be important to consider the effect of the winner's curse problem and, in particular, how it affects the bidding systems.

A second key contribution of our work is the introduction of our contextual second-price auction. While there are other known mechanisms that successfully help to mitigate the winner's curse, such as ascending-type auctions, they often require a dynamic sequence of bid elicitation. As such, they require a non-trivial communication cost between the auctioneer and the bidders, which might make those dynamic auctions less suitable for high frequency ad auctions. Our CSPA mechanism is static and require, at most, a single stage of communication between the auctioneer and the bidders making it more scalable in the ad auction setting.

From a technical standpoint, a key contribution of our work is the introduction of a mechanism design approach to address the effects of uncertainty and variance in machine learning prediction models on downstream selection mechanisms. In particular, ML techniques such as differential privacy and anonymization (see, for e.g. [14] for an application to ad modeling), which are often employed in practice, often inject noise in the signals they produce. Such signals are often used for selection and decision-making, opening the door for the systematic introduction of selection bias.[5] Our research opens a new avenue for alleviating sample bias (and as a consequence, prediction errors) by focusing on the design of mechanisms that operate in conjunction with these models.

Finally, we provide a novel contribution to the research literature on interdependent-value auctions by showing results robust to a general type of distributions. For profit-maximizing bidders, since the problem of characterizing the bidding equilibrium in the presence of interdependent value is challenging, most of the literature has restricted to the case where signals come from a distribution that is affiliated (for a textbook treatment on this subject, see Krishna [18, Chapters 6–11]). Notably, our technical results in the autobidding framework do not rely on such assumption. In fact, our results hold for any distribution. That technical tractability in the autobidding setting and the importance of autobidding in the ad-auctions industry makes our model particularly attractive for researchers and we expect that this line of work will grow.

## 2 Related Work

In the profit-maximizing setting, the seminal work of Milgrom and Weber [24] show that if bidders' signals on their values come from a distribution that satisfy the "affiliation property", then using an open auction, such as an English auction, can generate strictly more revenue than using a sealed-bid auction such as a second-price and first-price auction. Eden et al. [16] takes a computer science approach to that problem and using a worst-case performance

---

[3] We further show that even if bidders are unconstrained and can use a non-uniform bidding policy, we still have PoA = ∞.

[4] This can be done in the autobidding setting since it is often assumed that the platform has access to an estimate of the autobidders' value [7, 11].

[5] Consider for example the problem of selecting the highest signal among $n$ unbiased signals. Then the selected largest signal will overestimate its true value: it is more likely to be selected when there is a positive noise on the prediction than when there is a negative one.

analysis, along the lines of our price of anarchy approach, quantify the welfare loss of truthful auction when profit-maximizing bidders have interdependent valuations.

In the specific context of winner's curse in ad-auctions, Abraham et al. [1] study how different information structures on the signals advertisers receive through cookies can impact the performance of SPA. Arnosti et al. [5] study the case where a subset of advertisers have a fully precise signal of their value (performance advertisers) while others are completely unaware (brand advertisers). They show that SPA has a low performance and show that, instead, a modified SPA where a brand advertiser's bid gets a lift factor $\alpha \geq 1$ has good performance guarantees.[6] Both papers assume that bidders are profit-maximizing, our work instead, departs from that literature as it focuses on the case where bidders are autobidders with return-on-spend objectives.

Two other related works are Lahaie and McAfee [19] and Bax et al. [9] which, in addition to Milgrom and Weber [24], give additional examples of how mechanism design can be utilized to mitigate the effect of noisy signals. Lahaie and McAfee [19] consider an environment in which uncertainty appears in the clickthrough rate estimation while Bax et al. [9] consider an environment where the auctioneer receives unbiased estimates of value along with the variances. In both of these works, they construct a modified estimator that returns a more efficient ranking than simply ranking by raw signals. We too take a Bayesian approach to uncertainty by assuming that the private signal that each bidder receives is only an unbiased estimate of their true value. Our contextual second-price auction has a similar effect, in that it restores the efficient ranking in the auction. Importantly, our estimator leverages all predictions available at query time (in this sense, contextual information). This allows the platform to construct the best posterior estimate and thus rank efficiently as well as generate additional revenue. We distinguish from this line of work in that our work focuses on value-maximizing agents instead of profit-maximizing agents.

There is also work on designing mechanisms where the prediction uncertainty appears in a bandit setting [6, 15, 17]. In these works, the key assumption is that there is a set of ads with a fixed but unknown quality signal. If an ad is shown then one receives bandit feedback about the quality signal. For example, if the quality signal is the clickthrough rate then showing then upon showing the ad, the auctioneer receives binary feedback about whether the ad is clicked which the auctioneer can then use to update their estimate of the clickthrough rates. Note, in particular, that these works generally maintain independent estimates for each of the bidders. There is no interaction between the estimates and thus no correlation between the signals. The goal of these works is to design mechanisms with "low regret" with respect to a benchmark where the quality signal is known exactly. In our model instead, we focus on the case where signals are correlated and hence there is a winner's curse problem. Also, we study the mechanism design problem given a static ML prediction.

A large literature studies the problem of estimating clickthrough rates and conversion rates in the online ad setting, such as [10, 22, 26, 27] and references therein. The focus on static learning of

conversion rates highlights that predictions from ANNs and Deep Learning systems are inherently noisy, which is the starting point of our analysis.

Finally, we connect our contribution to the growing literature on autobidders in online ad auctions [2–4, 12, 13, 20, 21]. We show that in scenarios where bidders' signals are correlated (and thus a winner's curse problem), implementing a contextual auction format that varies across queries simplifies the problem faced by the autobidder, guarantees the optimality of uniform bidding even in the presence of noise in the platform's estimates, and provide better efficiency guarantees.

# 3  Model

We consider a setting with $n$ bidders who participate in ad auctions on an online platform. The set of single-slot queries follow a Bayesian model where the measure (or, probability) of queries where bidders valuations are $\mathbf{v} = (v_1, \ldots, v_n)$ comes from a distribution $F$ which could be correlated across bidders.

*Value Uncertainty.* We depart from the canonical autobidding model and assume that valuations $v_i$ are unknown to all agents (bidders and platform). Each bidder $i$ observes a signal $\tilde{v}_i$ which is an unbiased estimate of $v_i$. Mathematically, we assume that signals $\tilde{\mathbf{v}} = (\tilde{v}_1, \ldots, \tilde{v}_n)$ are drawn from a distribution $\tilde{F}$ and satisfy that:

(1) **Unbiased Estimator:** $\mathbb{E}[v_i | \tilde{v}_i] = \tilde{v}_i$.

(2) **Interdependence:** $\mathbb{E}[v_i | \tilde{\mathbf{v}}]$ may not be equal to $\mathbb{E}[v_i | \tilde{v}_i]$.

If $\mathbb{E}[v_i | \tilde{\mathbf{v}}] = \mathbb{E}[v_i | \tilde{v}_i] = \tilde{v}_i$ for all $\tilde{\mathbf{v}}$ then we say that the values have no interdependence.

Following a standard assumption in the autobidding literature, we assume that the prediction signal $\tilde{v}_i$ is known to the respective bidder and to the platform (see, for example [7, 8, 11]).

*Autobidding Problem.* We formulate the autobidding problem in our setting which adapts the classic model (see [3] for a comprehensive survey) to the setting where there may be interdependency among the bidders' values.

At a high level, the auto-bidding problem is to maximize the bidder's expected value subject to the constraint that their expected spend does not exceed the expected value. However, because other bidders' bids depend on their own private signals, which in turn, impacts the winner' assessment on its valuation, the optimal bidding decision now also depends on how other bidders are using their own signals at bidding time.

Let $\mathcal{I}_i$ denote a function which, given input $\tilde{\mathbf{v}}$, represents the information $\mathcal{I}_i(\tilde{\mathbf{v}})$ that bidder $i$ has at the time of bidding. A natural example of this is $\mathcal{I}_i(\tilde{\mathbf{v}}) = \tilde{v}_i$ which means that bidder $i$ only receives their own signal as information. However, we allow this function to be arbitrary: for example, it could be the identity function. Given $\mathcal{I}_i(\tilde{\mathbf{v}})$, the bidder needs to place a bid $b_i$ which denotes the maximum the bidder is willing to pay given the information it has. Now on every query, there is a minimum cost $c_i$ for the query (note that $c_i$ can still be random given $\tilde{\mathbf{v}}$; for example, the platform may still have a reserve price that is independent of $\tilde{\mathbf{v}}$). We assume that the auction satisfies the property that if bidder $i$ bids $b_i \geq c_i$ then bidder $i$ wins the query. Finally, if bidder $i$ wins the query then it has to pay a cost which we denote by $C_i \in [c_i, b_i]$. For truthful auctions, we have $C_i = c_i$, the minimum bid that it could have placed to win

---

[6]While Arnosti et al. [5] does not assume affiliation, they do require a heavy-tailed property on the distributions.

the auction. For FPA, we have $C_i = b_i$, the bidder's bid itself. Note that $c_i$ may not be revealed in some auction formats, such as FPA. The goal of the bidder is to choose a bidding function $b_i(\mathcal{I}_i(\tilde{\mathbf{v}}))$ to maximize its value subject to its ROS constraint.

The bidding problem can also be restated as follows which is more convenient from a technical point of view and is also consistent with prior work. We let $x_i(\mathcal{I}_i(\tilde{\mathbf{v}}), c_i)$ be bidder $i$'s decision of whether to buy a query where $c_i$ denotes the minimum possible cost that must be paid to win the query. This means that if the bidder places a bid $b_i$ then $x_i(\mathcal{I}_i(\tilde{\mathbf{v}}), c_i) = \mathbf{1}[c_i \leq b_i]$. Conversely, if $x_i(\mathcal{I}_i(\tilde{\mathbf{v}}), c_i)$ is decreasing in $c_i$ then we can recover the bid as $b_i = \sup\{c_i \,:\, x_i(\mathcal{I}_i(\tilde{\mathbf{v}}), c_i) = 1\}$. Finally, let $\mathcal{F}_i$ denote the $\sigma$-algebra generated by the outcome of the auction that is revealed to bidder $i$, particularly the value of $x_i(\mathcal{I}_i(\tilde{\mathbf{v}}), c_i)$ and $C_i$ if it won. For example, in both SPA and FPA, $\mathcal{F}_i$ reveals a lower bound on the winning bid if the bidder lost. On the other hand, if the bidder won, then in SPA $\mathcal{F}_i$ reveals the second-highest bid while in FPA $\mathcal{F}_i$ reveals only an upper bound on all other bids. The expected value that bidder $i$ gets under $x_i$ is given by

$$\mathbb{E}_{\tilde{\mathbf{v}}}[\mathbb{E}_{\mathbf{v}}[v_i|\mathcal{F}_i] \cdot x_i(\mathcal{I}_i(\tilde{\mathbf{v}}), c_i)]$$
$$= \mathbb{E}_{\tilde{\mathbf{v}}}[\mathbb{E}_{\mathbf{v}}[v_i \cdot x_i(\mathcal{I}_i(\tilde{\mathbf{v}}), c_i)|\mathcal{F}_i]]$$
$$= \mathbb{E}_{\tilde{\mathbf{v}}, \mathbf{v}}[v_i \cdot x_i(\mathcal{I}_i(\tilde{\mathbf{v}}), c_i)].$$

The first equality holds because the value of $x_i(\tilde{v}_i, c_i)$ is measurable under $\mathcal{F}_i$, while the second equality is a direct consequence of the law of iterated expectations. Similarly, the expected cost the bidder pays is given by $\mathbb{E}_{\tilde{\mathbf{v}}, \mathbf{v}}[C_i \cdot x_i(\mathcal{I}_i(\tilde{\mathbf{v}}), c_i)]$.

The autobidding problem that each bidder solves can be written as follows, where $x_i$ is the decision function:

$$\begin{aligned}
\text{maximize} \quad & \mathbb{E}_{\mathbf{v}, \tilde{\mathbf{v}}}[x_i(\mathcal{I}_i(\tilde{\mathbf{v}}), c_i) \cdot v_i] \\
\text{subject to} \quad & \mathbb{E}_{\mathbf{v}, \tilde{\mathbf{v}}}[x_i(\mathcal{I}_i(\tilde{\mathbf{v}}), c_i) \cdot (v_i - C_i)] \geq 0 \qquad \text{(ROS)} \\
& x_i(\mathcal{I}_i(\tilde{\mathbf{v}}), c_i) \text{ is non-increasing in } c_i.
\end{aligned}$$

At this point, we remind the reader that for truthful auctions, we have $C_i = c_i$ which is the case for some of the bidding results in prove in this paper.

*Uniform bidding.* Some of our results focus on the setting where bidders bid uniformly. In this paper, when we say uniform bidding, it will mean one of the following two options, which should be clear from context. Either $b_i(\mathcal{I}_i(\tilde{\mathbf{v}})) = \alpha_i \cdot \tilde{v}_i$ or $b_i(\mathcal{I}_i(\tilde{\mathbf{v}})) = \alpha_i \cdot \mathbb{E}[v_i|\tilde{\mathbf{v}}]$ for some constant $\alpha_i > 0$. Using the notation from above, this means that $x_i(\mathcal{I}_i(\tilde{\mathbf{v}}), c_i) = \mathbf{1}[\alpha_i \tilde{v}_i \leq c_i]$ or $x_i(\mathcal{I}_i(\tilde{\mathbf{v}}), c_i) = \mathbf{1}[\alpha_i \mathbb{E}[v_i|\tilde{\mathbf{v}}] \leq c_i]$. For our results regarding non-contextual auctions, such as SPA and FPA, this means the former since the bidder only has access to $\tilde{v}_i$. For our results regarding contextual auctions, uniform bidding refers to the latter since the bidder has access to $\mathbb{E}[v_i|\tilde{\mathbf{v}}]$. Due to its simplicity, it has been widely studied in the literature (see [3]).

*Equilibrium and Price of Anarchy.* Given an auction, a set of bidding functions $\mathbf{b}(\mathcal{I}_1(\tilde{\mathbf{v}}), \ldots, \mathcal{I}_n(\tilde{\mathbf{v}})) = (b_1(\mathcal{I}_1(\tilde{\mathbf{v}})), \ldots, b_n(\mathcal{I}_n(\tilde{\mathbf{v}})))$ forms an equilibrium if all the bidders satisfy their ROS constraint and no bidder can deviate to increase the value they receive without violating their autobidding ROS constraint. More specifically, let $x^{\mathbf{b}} : \mathbb{R}^n \to \{0, 1\}^n$ be the resulting allocation function given the bidding function $\mathbf{b}$ (we drop the dependence on $\mathcal{I}_i$ for notation).

In an equilibrium, we have that for any $i \in [n]$, for any bidding function $b'_i$, if $\mathbf{b}' = (b'_i, b_{-i})$ then either:

(1) $\mathbb{E}[x_i^{\mathbf{b}'}(\tilde{\mathbf{v}})(v_i - C_i)] < 0$; or
(2) $\mathbb{E}[x_i^{\mathbf{b}'}(\tilde{\mathbf{v}})v_i] < \mathbb{E}[x_i^{\mathbf{b}}(\tilde{\mathbf{v}})v_i]$.

To quantify the efficiency of different auctions, we make use of two standard metrics in the autobidding literature: the *liquid welfare* and the *price of anarchy*. For an allocation $x : \mathbb{R}^n \to \{0, 1\}^n$, the liquid welfare is defined as

$$\mathbb{E}_{\mathbf{v}, \tilde{\mathbf{v}}}\left[\sum_{i=1}^{n} x_i(\tilde{\mathbf{v}})v_i\right]. \qquad (3.1)$$

The *optimal liquid welfare* is obtained by choosing $x$ to optimize Eq. (3.1). Note that we have

$$\mathbb{E}_{\mathbf{v}, \tilde{\mathbf{v}}}\left[\sum_{i=1}^{n} x_i(\tilde{\mathbf{v}})v_i\right] = \mathbb{E}_{\tilde{\mathbf{v}}}\left[\sum_{i=1}^{n} x_i(\tilde{\mathbf{v}}) \cdot \mathbb{E}_{\mathbf{v}}[v_i|\tilde{\mathbf{v}}]\right],$$

so to optimize Eq. (3.1), we define the allocation $x$ as follows. For every $\tilde{\mathbf{v}}$, arbitrarily choose $i^* \in \arg\max_i \mathbb{E}[v_i|\tilde{\mathbf{v}}]$ and set $x_i(\tilde{\mathbf{v}}) = \mathbf{1}[i = i^*]$. We thus define $\text{OPT} = \mathbb{E}_{\tilde{\mathbf{v}}}[\max_i \mathbb{E}_{\mathbf{v}}[v_i|\tilde{\mathbf{v}}]]$. Note that OPT is defined with knowledge of all bidders' estimates and then choosing the optimal allocation.

Finally, the *price of anarchy* (PoA) of an auction is defined as the worst-case ratio of the liquid welfare between the auction and OPT (over all distributions $F$ and $\tilde{F}$).

## 4 Special Case: No Interdependence Between Values

In this section, we look at the special case where there is no interdependence between the bidders' values. More specifically, we assume that $\mathbb{E}[v_i|\tilde{\mathbf{v}}] = \mathbb{E}[v_i|\tilde{v}_i] = \tilde{v}_i$ for all $i \in [n]$. In other words, the signals of the other bidders do not provide any benefit in improving a bidder's value signal. This case essentially reduces to the private value setting that is now common in the autobidding literature [3]. However, for completeness, we give proofs for most of the results in this section.

The following theorem shows that if the auction is truthful than a best response for the bidder is to bid uniformly. Recall that bidding uniformly means $b_i(\mathcal{I}_i(\tilde{\mathbf{v}})) = \alpha_i \cdot \tilde{v}_i$ or, equivalently, $x_i(\mathcal{I}_i(\tilde{\mathbf{v}}), c_i) = \mathbf{1}[\alpha_i \cdot \tilde{v}_i \leq c_i]$ for some constant $\alpha_i > 0$. Also recall that for truthful auctions, we have $C_i = c_i$ which slightly simplifies Eq. (ROS).

**Theorem 4.1.** *Suppose that values are independent. Suppose that the auction is truthful and also that there exists a constant $\lambda > 0$ such that $\mathbb{E}[\mathbf{1}[\tilde{v}_i/c_i \geq \lambda/(1 + \lambda)] \cdot (\tilde{v}_i - c_i)] = 0$.[7] Then uniform bidding is an optimal bidding strategy. This is true even if each bidder has full access to $\tilde{\mathbf{v}}$.*

For the proof, we assume that $\mathcal{I}_i(\tilde{\mathbf{v}}) = \tilde{\mathbf{v}}$ and so we drop the $\mathcal{I}_i$. Eventually, we show that the bidder only uses $\tilde{v}_i$ despite having access to $\tilde{\mathbf{v}}$, as stated in the theorem.

---

[7] A sufficient condition for this assumption to hold is to assume that as we vary the bid multiplier $\alpha_i$ from 0 to $\infty$, both the value and cost increase continuously and the marginal ROI (defined as the ratio of the change in value with respect to an infinitesimal change in cost) is decreasing.

Proof. Let $\lambda$ denote the dual variable for the constraint in (ROS). The Lagrangian problem is to find $x_i$ to maximize

$$\mathcal{L}(x, \lambda) = \mathbb{E}[x_i(\tilde{\mathbf{v}}, c_i) \cdot \tilde{v}_i] + \lambda \cdot \mathbb{E}_v[x_i(\tilde{\mathbf{v}}, c_i) \cdot (\tilde{v}_i - c_i)] \quad (4.1)$$

subject to    $x_i(\tilde{\mathbf{v}}, c_i)$ is non-increasing in $c_i$.

An optimizer for $\mathcal{L}$ over $x_i$, for fixed $\lambda$, is $x_i^\lambda(\tilde{\mathbf{v}}, c_i) = \mathbf{1}[\tilde{v}_i/c_i \geq \lambda/(1+\lambda)]$. Note that with this choice of $x_i$, the constraint that $x_i(\tilde{\mathbf{v}}, c_i)$ is non-increasing in $c_i$ is satisfied. By standard weak duality, we have that (ROS) is upper bounded by $\mathcal{L}(x_i^\lambda, \lambda)$ for all $\lambda > 0$. Finally, choose $\lambda^*$ to satisfy $\mathbb{E}[\mathbf{1}[\tilde{v}_i/c_i \geq \lambda^*/(1+\lambda^*)] \cdot (\tilde{v}_i - c_i)] = 0$; the existence of such a value is guaranteed by the assumption of the theorem. Taking $x_i^{\lambda^*}$ as the solution to (ROS) shows that (ROS) is equal to $\mathcal{L}(x_i^{\lambda^*}, \lambda^*)$.

Now, note that $x_i^\lambda(\tilde{\mathbf{v}}, c_i) = \mathbf{1}[\tilde{v}_i/c_i \geq \lambda/(1+\lambda)]$ which implies that (i) uniform bidding is an optimal bidding strategy, by definition of uniform bidding, and (ii) this is the optimal bidding strategy even if the bidder has access to all of $\tilde{\mathbf{v}}$.                    □

The next theorem gives a PoA result when the values are not interdependent. Note that the theorem assumes that bidders always bid at least their value. First, we remark that it is straightforward to check that it is always a best-response for a bidder to bid at least their value. Second, without such an assumption, the PoA can be arbitrarily poor; for example, an equilibrium is for a low value bidder to always bid infinity while the remaining bidders bid 0.

**Theorem 4.2.** *The PoA of SPA is 2 provided that $b_i(\mathcal{I}_i(\tilde{\mathbf{v}})) \geq \tilde{v}_i$ for all $\tilde{\mathbf{v}}$ and $i$.*

The proof is essentially a simplified version (albeit in the Bayesian setting) of the proof that appears in [2].

Proof. Let $x^{\text{OPT}}(\tilde{v})$ denote any optimal allocation and $x^{\text{SPA}}(\tilde{v})$ denote an equilibrium allocation in SPA. Consider the two events

$$\mathcal{E}_1 = \{\tilde{v} : x^{\text{OPT}}(\tilde{v}) = x^{\text{SPA}}(\tilde{v})\}$$

and

$$\mathcal{E}_2 = \{\tilde{v} : x^{\text{OPT}}(\tilde{v}) \neq x^{\text{SPA}}(\tilde{v})\}.$$

On the event $\mathcal{E}_1$, we have that

$$\mathbb{E}_{\tilde{v}}\left[\sum_i v_i' \cdot x_i^{\text{SPA}}(\tilde{v})\mathbf{1}[\mathcal{E}_1]\right] = \mathbb{E}_{\tilde{v}}\left[\sum_i v_i' \cdot x_i^{\text{OPT}}(\tilde{v})\mathbf{1}[\mathcal{E}_1]\right]. \quad (4.2)$$

On the event $\mathcal{E}_2$, we have that the cost is at least $\sum_i v_i' \cdot x_i^{\text{OPT}}$ since the bid multiplier of every bidder is at least 1 and that bidder did not win. Thus, the ROS constraint implies that

$$\mathbb{E}_{\tilde{v}}\left[\sum_i v_i' \cdot x_i^{\text{SPA}}(\tilde{v})\right] \geq E_{\tilde{v}}\left[\sum_i v_i' \cdot x_i^{\text{OPT}}(\tilde{v})\mathbf{1}[\mathcal{E}_2]\right]. \quad (4.3)$$

Combining Eq. (4.2) and Eq. (4.3) gives that

$$2 \cdot \mathbb{E}_{\tilde{v}}\left[\sum_i v_i' \cdot x_i^{\text{SPA}}(\tilde{v})\right] \geq E_{\tilde{v}}\left[\sum_i v_i' \cdot x_i^{\text{OPT}}(\tilde{v})\right],$$

Which proves the upper bound of 2.

For the lower bound, consider the following example with two bidders. With probability 1/2, bidder 1 (resp. 2) has value 1 (resp. 0) for the query and with probability 1/2, bidder 1 (resp. 2) has value $\varepsilon$ (resp. 1) for the query. In this case, we claim that multipliers $\alpha_1 = 2/\varepsilon$ and $\alpha_2 = 1$ form an equilibrium. In this case, bidder 1

always wins for an average value of $(1 + \varepsilon)/2$ and an average cost of 1/2. Bidder 2 cannot deviate since if it does, it will get an average value of 1/2 but an average cost of 1.                    □

Next, a well-known result is that in FPA, if all bidders employ a uniform bidding strategy then optimal efficiency is always achieved.

**Theorem 4.3.** *In FPA, if the bidders use a uniform bidding strategy then a bid multiplier of 1 is an optimal multiplier and this gives PoA = 1.*

For a proof of this result, see [11, Theorem 6.5]. Note that a uniform bidding strategy may not be optimal under FPA. If $\mathcal{I}_i(\tilde{\mathbf{v}}) = \tilde{\mathbf{v}}$ and $b_i(\tilde{\mathbf{v}})$ can be an arbitrary function then the result of [20, Theorem 3.4] can be extended to the Bayesian setting to prove a PoA of 2.

## 5 Inefficiency of SPA and FPA with Interdependent Values

This section shows that when the values of the agents are interdependent, the standard second- and first-price auctions can result in poor outcomes. In Subsection 5.1, we show that uniform bidding is no longer an optimal bidding strategy which contrasts with Theorem 4.1. Next, in Subsection 5.2, we show that the price of anarchy can be arbitrarily bad when using these standard auctions.

### 5.1 Uniform Bidding is Suboptimal

In this section, we show that uniform bidding is suboptimal when the bidders receive only their own signal for the value of the query. In fact, the proof shows that uniform bidding can result in 0 value for the bidder.

**Theorem 5.1.** *Suppose that the values may be interdependent and that $\mathcal{I}_i(\tilde{\mathbf{v}}) = \tilde{v}_i$. Then there is an instance where uniform bidding in SPA is suboptimal for the bidder. In fact, the bidder can get* zero *value while satisfying its ROS constraint.*

At a high-level, the proof constructs two different types of queries. For one type of query, the bidder can get relatively cheaply relative to its *signal* but yet it knows that whenever it wins, it must overpay. It turns out that one can still design the signal so that it remains unbiased even in this case. The second type of query is more expensive, again relative to its signal but are queries which the bidder knows it will not overpay. A bidder that uses uniform bidding, even in a truthful auction, will not be able to target the second type of query.

Proof. For the proof, we focus on the problem for a single bidder and assume that the cost they need to pay is correlated with their true value. This can be because the cost is determined by another bidder whose value is correlated with the bidder we are looking at. Thus, for the proof, we drop the subscript $i$.

Consider the following example.

- (Query type A) With probability 1/2, we have $\tilde{v} \sim U(0, 0.5)$. Conditioned on this event:
  - (Query type A1) With probability 1/2, $v = \frac{1}{2}\tilde{v}$ and $c = 0.6\tilde{v}$.
  - (Query type A2) With probability 1/2, $v = \frac{3}{2}\tilde{v}$ and $c = 1.6\tilde{v}$.
- (Query type B) With probability 1/2, we have $\tilde{v} \sim U(0.5, 1)$ and $c = v = \tilde{v}$.

Note that $\mathbb{E}[v|\tilde{v}] = \tilde{v}$. For query type B, this is trivial since $v = \tilde{v}$. For query type A, this is because $v = \frac{1}{2}\tilde{v}$ and $v = \frac{3}{2}\tilde{v}$ with equal probability. Since the ranges of $\tilde{v}$ are disjoint in the two cases, it is straightforward to see that $\mathbb{E}[v|\tilde{v}] = \tilde{v}$ holds overall.

In the above example, note that no bid multiplier can guarantee positive value while the ROS constraint is met. To see this, observe that:

- A bid multiplier $\alpha < 0.6$, which results in a bid less than $0.6\tilde{v}$, wins none of the queries since the cost of all queries is at least $0.6\tilde{v}$.
- A bid multiplier $\alpha \in (0.6, 1)$, which results in a bid between $0.6\tilde{v}$ and $\tilde{v}$, wins query type A1. However, this means the ROS constraint is violated since $\mathbb{E}[(v-c)|\tilde{v}, \alpha\tilde{v} > c] = \mathbb{E}[(v-c)|\tilde{v}, c = 0.6\tilde{v}] = 0.5\tilde{v} - 0.6\tilde{v} = -0.1\tilde{v}$.
- A bid multiplier $\alpha \in (1, 1.6)$ wins query type A1 and B. Again, the ROS constraint is violated because winning query type A1 contributes $-0.1\tilde{v}$ to the ROS constraint while winning query type B contributes 0 to the ROS constraint.
- Finally, a bid multiplier $\alpha > 1.6$ wins all query types but query type A1 and A2 both negatively contribute to the ROS constraint.

On the other hand, an optimal, albeit non-uniform, bidding strategy is $b(\tilde{v}) = 1.01\tilde{v}\mathbf{1}[\tilde{v} > 0.5]$. In this case, the bidder wins whenever the query type is B and obtains a value of $\mathbb{E}_{\tilde{v}}[\mathbb{E}_v[v|\tilde{v} \in (0.5, 1)]\Pr[\tilde{v} \in (0.5, 1)]] = 3/8$. It is also straightforward to see that its ROS constraint is also met since the cost is always equal to its value. □

The issue with the example in the above proof is due to the correlation between a bidder's bid and the cost, which is a result of other bidders' value. If the bidder could somehow update its posterior based on other bidders' signal then note that a uniform bid multiplier of 1 does work. In that case, it would win only query type B and lose query type A1 and A2 because the cost on those queries is strictly larger than its value. This is exactly the contextual second-price auction we introduce in Section 6.

Our next theorem shows that interdependent values lead to new dynamics even in FPA when bidders bid uniformly. Well-known results show that the optimal bid multiplier should be 1 (Theorem 4.3) and this leads to an optimal allocation. However, as the next theorem shows, in the interdependent setting, the bid multiplier may be less than 1 and ultimately, this leads to inefficiency.

**Theorem 5.2.** *Suppose that bidders only use uniform bidding strategies. When values are interdependent, the optimal bid-shading constant may be less than* 1.

Proof. Suppose there are two bidders. The value of bidder 1 is $v_1 \sim U(0, 1)$. However, bidder 1 only receives $\tilde{v}_1 = \mathbb{E}[v_1] = 1/2$ as the signal. Clearly, $\mathbb{E}[v_1|\tilde{v}_1] = \tilde{v}_1$ since the signal reveals no information (other than the mean). Let $\beta > 1$ be a constant. The value of bidder 2 is $v_2 = v_1^{\beta}$ and bidder 2 receives $\tilde{v}_2 = v_2$ as the signal. Let us assume bidder 2 uses a bid multiplier of 1, which in fact is the optimal bid multiplier. We will show that bidder 1 should always bid shade.

Suppose that bidder 1 bids $b \leq 1$ (it has no reason to bid more than 1 since otherwise it can decrease its bid to 1 without affecting

its allocation). Then its expected value is

$$\mathbb{E}[v_1|v_2 < b] \cdot \Pr[v_2 < b]$$

$$= \mathbb{E}[v_1|v_1 < b^{1/\beta}] \cdot \Pr[v_1 < b^{1/\beta}] = \frac{b^{2/\beta}}{2}. \quad (5.1)$$

Its expected cost is

$$b \cdot \Pr[v_2 < b] = b \cdot \Pr[v_1 < b^{1/\beta}] = b^{1+1/\beta}. \quad (5.2)$$

For a given bid $b$, let $f(b) = \frac{b^{2/\beta}}{2} - b^{1+1/\beta}$ be the slack in the ROS constraint. Differentiating, we have

$$f'(b) = \frac{b^{1/\beta}(b^{1/\beta-1} - (\beta+1))}{\beta}.$$

The key observation is that the sign of $f'$ is determined by $b^{1/\beta-1} - (\beta+1)$, this is decreasing in $b$, and is positive for $b$ sufficiently small. Since $f(0) = 0$ and $f(1) = -1/2$, we conclude that $f(b) > 0$ when $b$ is sufficiently small and $f$ has exactly one root in $(0, 1)$. Since the value is an increasing function of $b$, this means that the optimal bid is precisely the root of $f$, i.e. when its ROS constraint is tight.

Thus, equating Eq. (5.1) and Eq. (5.2), we have that bidder 1's optimal bid is $b = 2^{-\beta/(\beta-1)} < 1/2$ since $\beta > 1$. We conclude that bidder 1's best response is to bid shade. □

Let us point out that for the example in the above proof, bidder 1 always has the higher value since $v_2 = v_1^{\beta} < v_1$ as $v_1 \in (0, 1)$ almost surely. In the proof, we did compute a set of equilibrium bid multipliers so this gives an example where bidders may use uniform bidding but the equilibrium allocation in FPA is not efficient for interdependent signals. In the next section, we give another example where uniform bidding leads to inefficient outcomes.

## 5.2 SPA and FPA have Infinite Price of Anarchy

In this section, we show that SPA and FPA can have arbitrarily poor PoA when the only signal they receive is $\tilde{v}_i$.

**Theorem 5.3.** *For every integer $k \geq 3$, there exists an instance where the price of anarchy is at least $k/2$ for both SPA and FPA when the bidders only receive $\tilde{v}_i$ as their signal.*

We give a high-level overview of the proof. Essentially, we construct an instance with $k + 2$ bidders where, with probability one, one of the first $k$ bidders has high value. However, as most of the time (with probability $1 - 1/k$), each bidder has low value, they are reluctant to bid high. There is a very low value bidder whose signal is correlated in a way that reveals the high value bidder. However, as their values are too low, their signal is essentially never revealed. Finally, there is one bidder whose value is slightly more than the expectation of the first $k$ bidders discussed above which always ends up winning.

Proof. The instance consists of $k + 2$ bidders. Consider the following correlated distribution among the bidders. Choose $i \in [k]$ uniformly at random. We assume (i) bidder $i$ has $v_i = k/2$ and $v_j = 0$ for $j \in [k] \setminus \{i\}$, (ii) bidder $k + 1$ has value $v_{k+1} = 1$, and (iii) bidder $k + 2$ has value $v_{k+2} = i/(100k)$. For signals, we assume that (i) each bidder $i \in [k]$ receives $\tilde{v}_i = 1/2$ and (ii) for the remaining $i \in \{k + 1, k + 2\}$, bidder $i$ receives $\tilde{v}_i = v_i$. It is straightforward to verify that for every $i \in [k + 2]$, we have $\mathbb{E}[v_i|\tilde{v}_i] = \tilde{v}_i$.

*Optimal welfare.* Recall that the optimal welfare is defined as $\mathbb{E}[\max_i \mathbb{E}[v_i|\tilde{\mathbf{v}}]]$. In this case $\max_i \mathbb{E}[v_i|\tilde{\mathbf{v}}] = k/2$ since given $\tilde{v}_{k+2}$, we can identify $i \in [k]$ such that $v_i = k/2$. Thus, the optimal welfare is $\mathbb{E}[\max_i \mathbb{E}[v_i|\tilde{\mathbf{v}}]] = k/2$.

*Equilibrium welfare.* We claim that all bidders using a bid multiplier of 1 is an equilibrium in both SPA and FPA. First, for each $i \in [k]$, bidder $i$ is facing a price of 1 which is due to $k + 1$. If bidder $i$ places a bid $b_i$ such that $b_i > 1$ then we have $\mathbb{E}[v_i|\tilde{v}_i, b_i > 1] = 0.5$ because the event $\{b_i > 1\}$ reveals no additional information on any of the other bidders. Thus, $\mathbb{E}[v_i - c_i|b_i > 1] \leq 0.5 - 1 < 0$. We conclude that bidder $i$ cannot profitably deviate without violating its ROS constraint. Second, bidder $k + 1$ will not deviate since it is always winning and at a price that is at most its own value.[8] Third, bidder $k + 2$ will not deviate since its value is strictly less than 1 and to win, it must pay at least 1. We thus conclude that the equilibrium welfare is 1.

To summarize, the optimal welfare is at least $k/2$ and we exhibited an equilibrium where the welfare is 1. This completes the proof. □

## 6 Contextual Second-Price Auction

We now define the Contextual Second-Price Auction. In Subsection 6.1, we show that under CSPA, uniform bidding becomes an optimal bidding strategy. In Subsection 6.2, we show that CSPA has a PoA of 2.

There are at least two different ways to implement a contextual auction. One possibility is to provide $\tilde{\mathbf{v}}$ to every bidder so that each bidder can compute $\mathbb{E}[v_i|\tilde{\mathbf{v}}]$.

Algorithm 1 provides an alternative auction to implement CSPA where all the corrections are done within the auction. Under this auction, the bidder places a uniform bid on $\tilde{v}_i$ (shown in Theorem 6.1). The key difference from SPA is that the auction computes an updated posterior based on the value estimates of all the participating advertisers and then uses this as a correction multiplier on the bidder's original bid. With this auction, we recover both properties that SPA has under non-interdependent values – optimality of uniform bidding and a PoA of 2 – even under interdependent values.

---

**Algorithm 1:** Contextual Second-Price Auction (CSPA)

**Input:** Bids $b_i$ and predicted values $\tilde{\mathbf{v}}$
**Output:** Winner's identity $W$ and its payment $p$
**for** $i \in [n]$ **do**
  Compute $v_i' \leftarrow \mathbb{E}_v[v_i|\tilde{\mathbf{v}}]$.
  Adjust bids: $b_i' \leftarrow \frac{v_i'}{\tilde{v}_i} b_i$.
**end**
Run SPA on updated bids:
  $W \leftarrow \operatorname{argmax}_i b_i', \ p \leftarrow \max_{j \neq i} b_j'$

---

Let us point out that the bidders do not actually need access to $\tilde{\mathbf{v}}$. In fact, as seen in Algorithm 1, the auction can implement

---

[8]While bidder $k + 1$ can lower its price, since there are no other queries available, it cannot increase its value. Since the bidder is a value-maximizer, this means that bidder $k + 1$ is indifferent between bidding 1 and bidding 0.51.

the correction on behalf of the bidder. Thus, the bidder may still use uniform bidding with respect to $\tilde{v}_i$ which the auction can then correct to a uniform bid on $\mathbb{E}[v_i|\tilde{\mathbf{v}}]$.

### 6.1 Uniform Bidding in Contextual Truthful Auction is Optimal

In this section, we show that uniform bidding is optimal when the underlying auction is a contextual truthful auction. By contextual, we essentially mean that $I_i(\tilde{\mathbf{v}}) = \tilde{\mathbf{v}}$ instead of $I_i(\tilde{\mathbf{v}}) = \tilde{v}_i$ as in the previous section. We let $\mathcal{F}$ denote the $\sigma$-algebra that represents the information the bidder receives after the outcome of the auction. In particular, this includes $\tilde{\mathbf{v}}$ (though the bidder has this information prior to the auction as well). In the case of CSPA, without a reserve, $\mathcal{F}$ is simply $\tilde{\mathbf{v}}$, assuming that the other bidders' bidding function is fixed. If there is a reserve then $\mathcal{F}$ includes the reserve price on top of $\tilde{\mathbf{v}}$. The bidding problem can then be written as

$$\begin{aligned}
\text{maximize} \quad & \mathbb{E}_{\tilde{\mathbf{v}}}[x_i(\tilde{\mathbf{v}}, c_i) \cdot v_i] \\
\text{subject to} \quad & \mathbb{E}_{\tilde{\mathbf{v}}}[x_i(\tilde{\mathbf{v}}, c_i) \cdot (v_i - c_i)] \geq 0 \\
& x_i(\tilde{\mathbf{v}}, c_i) \text{ is non-increasing in } c_i.
\end{aligned} \quad (6.1)$$

Recall here that $c_i$ is the minimum bid needed to win the auction which, for truthful auctions, is equal to the cost. Let us also recall that the function $x(\cdot, \cdot)$ is known to bidder; however, its realization is random as its inputs are random.

**Theorem 6.1.** *For contextual and deterministic truthful auctions, if there exists $\lambda > 0$ such that*

$$\mathbb{E}[\mathbf{1}[\mathbb{E}[v_i|\tilde{\mathbf{v}}]/c_i \geq \lambda/(1+\lambda)] \cdot (\mathbb{E}[v_i|\tilde{\mathbf{v}}] - c_i)] = 0 \quad (6.2)$$

*then uniform bidding is an optimal bidding strategy.*

Proof. Taking the Lagrangian of Eq. (6.1), we have

$$\begin{aligned}
\mathcal{L}(x_i, \lambda) &= \mathbb{E}[x_i(\tilde{\mathbf{v}}, c_i) \cdot ((1+\lambda)v_i - \lambda c_i)] \\
&= \mathbb{E}[\mathbb{E}[x_i(\tilde{\mathbf{v}}, c_i) \cdot ((1+\lambda)v_i - \lambda c)|\mathcal{F}]] \\
&= \mathbb{E}[x_i(\tilde{\mathbf{v}}, c) \cdot \mathbb{E}[(1+\lambda)v_i - \lambda c_i)|\mathcal{F}]] \\
&= \mathbb{E}[x(\tilde{\mathbf{v}}, c_i) \cdot ((1+\lambda)\mathbb{E}[v_i|\tilde{\mathbf{v}}] - \lambda c_i)].
\end{aligned}$$

Note that the third equality is because the value of $x_i(\tilde{\mathbf{v}}, c)$ is in $\mathcal{F}$ so it can be taken out of the conditional expectation. To optimize $\mathcal{L}(x_i, \lambda)$ for a fixed $\lambda$, we can take $x_i(\tilde{\mathbf{v}}, c_i) = \mathbf{1}[(1+\lambda)\mathbb{E}[v_i|\tilde{\mathbf{v}}] \geq \lambda c_i]$. Note that this satisfies the last constraint in Eq. (6.1) which is necessary since it is not included in the Lagrangian. As discussed in Section 3, this is the definition of a uniform bidding strategy.

Finally, if we take $\lambda$ to satisfy Eq. (6.2), i.e. $x_i(\tilde{\mathbf{v}}, c_i) = \mathbf{1}[(1+\lambda)\mathbb{E}[v_i|\tilde{\mathbf{v}}] \geq \lambda c_i]$ then we have (i) $x_i$ optimizes $\mathcal{L}(x_i, \lambda)$, as discussed in the previous paragraph, and the value is $\mathbb{E}[x_i(\tilde{\mathbf{v}}, c_i) \cdot v_i]$ and (ii) Eq. (6.1) also has value $\mathbb{E}[x_i(\tilde{\mathbf{v}}, c_i) \cdot v_i]$. We thus conclude that uniform bidding is an optimal bidding strategy. □

### 6.2 Efficiency of CSPA

This theorem proves two theorems on the efficiency of CSPA. First, we show that CSPA has a PoA of 2 which recovers the efficiency guarantee of SPA in the independent value setting. Second, we show that there are instances where SPA actually performs better than CSPA but that the gap is at most 2.

**Theorem 6.2.** *Assume that bidders bid uniformly and that every bidder uses a bid multiplier $\alpha_i \geq 1$. Then the PoA of CSPA is equal to 2.*

We only prove the upper bound on the PoA. The lower bound of 2 is inherited from Theorem 4.2 since the setting without any interdependence between the values is a special case.

PROOF. Recall that each bidder $i$ chooses a bid multiplier $\alpha_i \geq 1$ and then places an initial bid $b_i = \alpha_i \cdot \tilde{v}_i$. The bid is then updated by multiplying by the ratio $\frac{\mathbb{E}[v_i|\tilde{v}]}{\tilde{v}_i}$ so that $b_i' = \alpha_i \cdot \mathbb{E}[v_i|\tilde{v}]$. For notation, let $v_i' = \mathbb{E}[v_i|\tilde{v}]$.

Let $x^{\text{OPT}}(\tilde{v})$ denote any optimal allocation and $x^{\text{CSPA}}(\tilde{v})$ denote an equilibrium allocation in CSPA. Consider the two events

$$\mathcal{E}_1 = \{\tilde{v} \, : \, x^{\text{OPT}}(\tilde{v}) = x^{\text{CSPA}}(\tilde{v})\}$$

and

$$\mathcal{E}_2 = \{\tilde{v} \, : \, x^{\text{OPT}}(\tilde{v}) \neq x^{\text{CSPA}}(\tilde{v})\}.$$

On the event $\mathcal{E}_1$, we have that

$$\mathbb{E}_{\tilde{v}}\left[\sum_i v_i' \cdot x_i^{\text{CSPA}}(\tilde{v})\mathbf{1}[\mathcal{E}_1]\right] = \mathbb{E}_{\tilde{v}}\left[\sum_i v_i' \cdot x_i^{\text{OPT}}(\tilde{v})\mathbf{1}[\mathcal{E}_1]\right]. \quad (6.3)$$

On the event $\mathcal{E}_2$, we have that the cost is at least $\sum_i v_i' \cdot x_i^{\text{OPT}}$ since the bid multiplier of every bidder is at least 1 and that bidder did not win. Thus, the ROS constraint implies that

$$\mathbb{E}_{\tilde{v}}\left[\sum_i v_i' \cdot x_i^{\text{CSPA}}(\tilde{v})\right] \geq E_{\tilde{v}}\left[\sum_i v_i' \cdot x_i^{\text{OPT}}(\tilde{v})\mathbf{1}[\mathcal{E}_2]\right]. \quad (6.4)$$

Combining Eq. (6.3) and Eq. (6.4) gives that

$$2 \cdot \mathbb{E}_{\tilde{v}}\left[\sum_i v_i' \cdot x_i^{\text{CSPA}}(\tilde{v})\right] \geq E_{\tilde{v}}\left[\sum_i v_i' \cdot x_i^{\text{OPT}}(\tilde{v})\right],$$

as desired. □

While CSPA improves upon SPA in the worst case, it is natural to ask whether or not CSPA is always better than SPA. It turns out the answer is no. The intuition is that there can be instances where the noise actually helps to prevent "inversions", where a bidder with higher value loses to another bidder with lower value because the lower value bidder is able to gain some slack in their ROS constraint on some other query. In such scenarios, SPA can actually be more efficient. However, the next theorem shows that the improvement of SPA over CSPA is fairly limited and the ratio can be at most a factor 2 apart.

To formally state the result, we need one last bit of notation. For an instance $I$, let $W_{\text{CSPA}}(I)$ (resp. $W_{\text{SPA}}(I)$) denote the worst-case liquid welfare for CSPA (resp. SPA) over all equilibria for the instance $I$.

**Theorem 6.3.** *For any instance $I$, we have $\frac{1}{2} \leq \frac{W_{CSPA}(I)}{W_{SPA}(I)} < \infty$. Moreover, neither inequality can be improved.*

PROOF. For the proof, we fix an instance $I$ and we drop the dependence on $I$. The fact that $W_{\text{CSPA}}/W_{\text{SPA}} < \infty$ is trivial as long as the instance $I$ has at least one agent with strictly positive value for at least one query. To see that this bound cannot be improved, note that Theorem 5.3 implies that for $k \geq 3$, there exists an instance $I$ such that the optimal allocation has value $k/2$ but $W_{\text{SPA}} \leq 1$. By

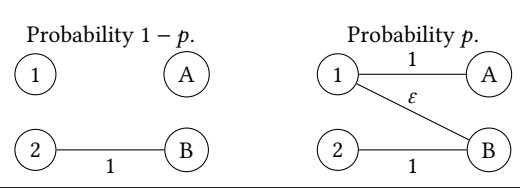

**Figure 1:** The instance where SPA improves on CSPA by a factor of 2. Numbers next to edges denote values; missing edges denote 0 value. First, we pick either the graph on the right or the graph on the left and then pick query $A$ or query $B$ uniformly at random. A third bidder determines which of the two graphs occurs.

Theorem 6.2, we have $W_{\text{CSPA}} \geq k/4$ so $W_{\text{CSPA}}/W_{\text{SPA}} \geq k/4$. As $k$ is arbitrary, this proves that the upper bound cannot be improved.

For the lower bound, the inequality $W_{\text{CSPA}}/W_{\text{SPA}} \geq 1/2$ is an implication of Theorem 6.2. It remains to show that this inequality is tight.

We consider the following instance with three bidders which is illustrated in Figure 1. Let $0 < \delta < \varepsilon < 1$ be parameters. Let $p = 1/(1 + 2\varepsilon)$. With probability $1 - p$, bidder 0 has value $v_0 = 0$ and with probability $p$ bidder 0 has value $v_0 = \delta$. Bidder 0 receives $\tilde{v}_0 = v_0$ as its signal. The value distribution for bidders 1 and 2 is defined conditional on the value for bidder 0. If $v_0 = 0$ then $(v_1, v_2) = (0, 0)$ with probability $1/2$ and $(v_1, v_2) = (0, 1)$ with probability $1/2$. If $v_0 = \delta$ then $(v_1, v_2) = (1, 0)$ with probability $1/2$ and $(v_1, v_2) = (\varepsilon, 1)$ with probability $1/2$. Bidder 1 gets $\tilde{v}_1 = \frac{p(1+\varepsilon)}{2}$ as its signal and bidder 2 gets $\tilde{v}_2 = v_2$ as its signal.

*Welfare in SPA.* Consider any equilibrium where every bidder uses a bid multiplier of at least 1. First, observe that bidder 0 never wins since their expected cost is at least $1/2$ (due to bidder 2 alone) while their value is at most $\delta/2$. Next, we claim that bidder 1 wins if $v_1 = 1$ and bidder 2 wins if $v_2 = 1$. Indeed, if bidder 2 does not when $v_2 = 1$ it must be that bidder 1 chooses a multiplier $\alpha_1$ such that $\alpha_1 \tilde{v}_1 > 1$ in which case bidder 1 wins every query. Its expected value is thus $\frac{p(1+\varepsilon)}{2} < 1/2$ while its expected cost is $1/2$. Thus bidder 1's ROS constraint is violated. So bidder 2 must win when $v_2 = 1$. When $v_1 = 1$ then we know that $v_2 = 0$ so in this case, bidder 1's bid is at least $\frac{p(1+\varepsilon)}{2} > \delta > 0$. So bidder 1 wins. The welfare in this case is at least $p$ (since when $v_0 = 0$, the higher bidder wins).

*Welfare in CSPA.* In CSPA, we claim that an equilibrium is for bidder 0 and 2 to use bid multipliers of 1 and for bidder 1 to use an arbitrarily large bid multiplier. Indeed, if $v_0 = 0$ then bidder 1 has value 0 and wins nothing while bidder 2 wins a query with value 1 and cost $\delta$ with probability $1/2$. On the other hand, if $v_0 = \delta$ then bidder 1 wins both queries at a total price of $(1 + \delta)/2$ and gets a total value of $(1 + \varepsilon)/2$. Bidder 2 has no incentive to deviate since bidder 1's bid is arbitrarily large. Thus, the welfare in CSPA is $p(1 + \varepsilon)/2 + (1 - p)/2$.

*Comparison of SPA and CSPA welfare.* Note that as $\varepsilon \to 0$, we have $p \to 1$. Thus, the welfare of SPA tends to at least 1 as $\varepsilon \to 0$ while the welfare of CSPA tends to $1/2$ as $\varepsilon \to 0$. We conclude that the inequality $W_{\text{CSPA}}/W_{\text{SPA}} \geq 1/2$ cannot be improved. □

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

Received 20 February 2007; revised 12 March 2009; accepted 5 June 2009

