# OpenReview forum: "Autobidding With Interdependent Values"
_ACM.org/TheWebConf/2025/Conference — WWW 2025 Oral_

### Official Review · Reviewer_meYd · 2024-11-20

**Novelty:** 3
**Technical Quality:** 2

**Review:**

The paper explores the challenges of autobidding in online advertising auctions where bidders' signals can be noisy and correlated, leading to the "winner's curse." The authors demonstrate the inefficiency of traditional auctions like the second-price auction (SPA) and the first-price auction (FPA) under these conditions. They propose a novel mechanism, the Contextual Second Price Auction (CSPA), which adjusts the allocation of SPA by leveraging multiple signals to mitigate adverse effects. The paper shows that uniform bidding is an optimal strategy in CSPA and proves a tight bound on the price of anarchy for CSPA of 2, aligning with results from independent settings. Additionally, CSPA is shown to achieve at least half the welfare of SPA.

Pros:
1. The paper provides a thorough theoretical analysis with clear definitions, theorems, and proofs that establish the properties of CSPA.

Cons:
1. The paper is highly technical and may be difficult for readers.
2. While the theoretical analysis is robust, the paper could benefit from empirical evaluations or simulations to demonstrate the practical implications of CSPA.
3. The paper makes several assumptions about the auction environment and bidder behavior that may not always hold in real-world scenarios. For example, the authors assume that bidders are rational and bidders have perfect information about the auction process.

**Questions:**

1. Is CSPA considered a value calibration algorithm? What distinguishes CSPA from other value calibration algorithms, and what are its advantages?
2. Is there any other algorithm that can substitute CSPA?
3. How does the CSPA algorithm perform in PoA compared to existing algorithms? Such as value calibration algorithms.

**Reviewer Confidence:**

3: The reviewer is confident but not certain that the evaluation is correct

**Scope:**

4: The work is relevant to the Web and to the track, and is of broad interest to the community

---

### Official Review · Reviewer_uEhL · 2024-11-30

**Novelty:** 4
**Technical Quality:** 5

**Review:**

The paper considers an auction (auto bidding) problem where agents receive noisy signals on their own values, which are interdependent. They show theoretical guarantees of equilibrium and POA under first-price auctions and second-price auctions when independent signals fail in the interdependent signal setting, and the worst case can be arbitrarily bad. To solve this problem, they propose a contextual auction mechanism where the mechanism helps adjust the information bias in the bidder's bids. They show that this contextual auction mechanism recovers the old theoretical guarantees under interdependent signals.

Pros:
1. The introduction is very well-written. It motivates the significance of considering interdependent values, explains their main contribution, and clearly sorts out the relationship between this paper and the existing literature.
2. The theoretical results are non-trivial, solid, and have a clear structure.

Weaknesses:
1. The contextual mechanism is assumed to know all the signals of all bidders. I feel this is way too strong an assumption. This means that the mechanism knows 100% what every bidder knows and can perfectly act as every bidder. The resemblance of this assumption under a non-Bayesian valuation is ''every bidder knows their own value, and the mechanism knows all the values''. In this way (both Bayesian and Non-Bayesian), I doubt the necessity to still hold an auction. The mechanisms can take no input and utilize all these signals to reach an outcome that fulfills some goal (max-revenue or max-welfare). I feel that a more reasonable assumption is that the mechanism knows the distribution or has some more coarse information about each value signal but not exactly the signal itself.
2. The paper is written in a way that readers are assumed to be somewhat familiar with the auto bidding problem. I'm not sure if it really counts as a ''weakness'', but I get confused many times in reading the paper when the setting is different from a traditional auction setting and unexplained.

**Questions:**

I am not familiar with auto bidding, so most questions are clarifications.

1. Please justify your assumption on the mechanisms knowing all the signals. By the way, I tried to look for similar assumptions in [7], [8], and [11] you mentioned but eventually got lost. Could you please point out where this assumption is located in these papers?
2. What is the main (high-level) difference between auto-bidding and traditional auctions, and how are these differences reflected in your modeling and goal of auto-bidding? (For example, are the ''queries'' equivalent to ''bidders''?)
3. Continue question on 2. The cost $c_i$ in your modeling is something that confuses me the most. A traditional auction mechanism has an allocation function $x_i$ and a payment function $p_i$ for each agent. But your paper models a cost $c_i$ for each agent, and $x_i$ becomes a function of $c_i$. Why does this modeling happen?
4. What does the ''truthful'' auction refer to in Theorem 4.1? I guess it refers to ''an auction mechanism where revenue maximizing agents are incentivized to bid truthfully'. Is that correct?
5. Line 503-504, what is $v_i'$?

**Reviewer Confidence:**

2: The reviewer is willing to defend the evaluation, but it is likely that the reviewer did not understand parts of the paper

**Scope:**

4: The work is relevant to the Web and to the track, and is of broad interest to the community

---

### Official Review · Reviewer_tw5L · 2024-12-02

**Novelty:** 6
**Technical Quality:** 5

**Review:**

# Paper summary

This paper examines the autobidding problem in scenarios where bidders have interdependent values. Consistent with the autobidding literature, it assumes that bidders are value maximizers subject to return-on-spend (ROS) constraints. However, it departs from previous works by considering settings where bidders are initially unaware of their exact values and instead rely on signals. These signals are assumed to be unbiased and interdependent.

The main results of the paper are as follows:

1. The authors show that interdependent signals cause the original constant price of anarchy (PoA) guarantee for both the first-price auction (FPA) and the second-price auction (SPA) to deteriorate to infinity [Theorem 5.2]. Additionally, uniform bidding is shown to be suboptimal in SPA [Theorem 5.1].

2. The authors propose the CSPA as a new generalization of SPA to the interdependent value model studied in this work. They establish that uniform bidding is optimal in this context [Theorem 6.1] and that the original PoA of 2 can be restored [Theorem 6.2].

3. The authors further discuss the efficiency of the SPA and CSPA by deriving the exact gap between them [Theorem 6.3].


# Evaluation

- This paper contributes to the rapidly growing literature on the autobidding. Studying the settings where the bidders have interdependent signals is theoretically reasonably and practically motivated.

- The paper provides a relatively comprehensive study. They first show the failure of the classic mechanisms (FPA, SPA), and then design a new mechanism (CSPA) and prove its tight PoA of 2. Overall, the results are solid and technically interesting. I enjoy reading this paper.

**Questions:**

Similar to the contextual second-price auction, it seems also natural to consider a contextual first-price auction (CFPA). Could you provide insights into this mechanism, particularly regarding its PoA guarantee?

**Reviewer Confidence:**

3: The reviewer is confident but not certain that the evaluation is correct

**Scope:**

4: The work is relevant to the Web and to the track, and is of broad interest to the community

---

### Official Review · Reviewer_LqTn · 2024-12-02

**Novelty:** 6
**Technical Quality:** 4

**Review:**

The paper at hand revisits the autobidder setting, but considers the case where bidder values are interdependent. The motivation is that different advertisers might see different signals of value, but the true value to any bidder (advertiser) may depend on all the signals. It proves a few results in this setting:

1. The uniform bid shading rule (which can be shown to be optimal in private value settings, 4.1/ 4.2), has curious properties in the interdependent setting (5.1).
2. The SPA and FPA have poor POA (5.3)
3. What they call a CSPA restores the nice properties of private value settings (6.1,2,3)

Overall I had a few questions, mostly about the interpretation of this model.

1. For the positive results of CSPA, what we need is an agent/ autobidder's ability to estimate the bidder's value from the profile of signals. While I do understand that conceptually there may be interdepedence, in reality these signals are abstract and correspond to agents' underlying data science/ ML. So no one knows the joint distribution, and estimating that might be much harder.
2. similarly, some of these may correspond to additional private signals (e.g. i purchased additional tracking data for bidders that you did not) or deeper ML by my team. It's not clear, therefore that CSPA will be a workable solution in practice since it potentially also leads to free-rider problems.
3. Minor: why is the last line of the document "Received 20 February..." on the references page?

**Questions:**

--

**Reviewer Confidence:**

3: The reviewer is confident but not certain that the evaluation is correct

**Scope:**

3: The work is somewhat relevant to the Web and to the track, and is of narrow interest to a sub-community

---

### Official Review · Reviewer_Eca1 · 2024-12-03

**Novelty:** 5
**Technical Quality:** 5

**Review:**

The paper investigates value interdependence in autobidding, where the valuation is unknown to the bidder, and a signal of the value can be observed and is correlated with other's value. The authors first showed that this noisy and interdependent signal setting will change the classic autobidding result: uniform bidding is no longer an optimal bidding strategy for SPA and both SPA and FPA can have arbitrarily large PoA. To mitigate the problem, the authors further proposed contextual SPA, where the auction have access to the noisy signal and can compute the unbiased estimator and adjust the bid on behalf of the bidder. The authors show that uniform bidding is an optimal bidding strategy in CSPA and that CSPA achieves at least half the welfare of SPA.

Quality and clarity: The paper is very well-written with good motivation and clearly differentiate the setting with the classic autobidding setting. The setting is interesting and realistic in online marketplace/advertising.

Significance: Though there are prior works on interdependent value in auctions, to my knowledge, this paper is novel in that it considers interdependent value in autobidding and proposed CSPA to mitigate winner's curse under controllable amount of communication with the bidders.

**Questions:**

Could the authors please comment on how the bidders / the auction may have access to the signals v tilde in real market / ad auctions? Is the signal coming from each bidder's individual predictor?

**Reviewer Confidence:**

2: The reviewer is willing to defend the evaluation, but it is likely that the reviewer did not understand parts of the paper

**Scope:**

4: The work is relevant to the Web and to the track, and is of broad interest to the community